# Roles of Estrogens in the Healthy and Diseased Oviparous Vertebrate Liver

**DOI:** 10.3390/metabo11080502

**Published:** 2021-07-30

**Authors:** Blandine Tramunt, Alexandra Montagner, Nguan Soon Tan, Pierre Gourdy, Hervé Rémignon, Walter Wahli

**Affiliations:** 1Institut des Maladies Métaboliques et Cardiovasculaires (I2MC-UMR1297), INSERM/UPS, Université de Toulouse, F-31432 Toulouse, France; blandine.tramunt@inserm.fr (B.T.); alexandra.montagner@inserm.fr (A.M.); pierre.gourdy@inserm.fr (P.G.); 2Service de Diabétologie, Maladies Métaboliques et Nutrition, CHU de Toulouse, F-31059 Toulouse, France; 3Lee Kong Chian School of Medicine, Nanyang Technological University Singapore, Clinical Sciences Building, Singapore 308232, Singapore; nstan@ntu.edu.sg; 4INP-ENSAT, Université de Toulouse, F-31320 Castanet-Tolosan, France; herve.remignon@toulouse-inp.fr; 5Toxalim Research Center in Food Toxicology (UMR 1331), INRAE, National Veterinary College of Toulouse (ENVT), Purpan College of Engineers of the Institut National Polytechnique de Toulouse (INP-PURPAN), Université Toulouse III—Paul Sabatier (UPS), Université de Toulouse, F-31300 Toulouse, France; 6Center for Integrative Genomics, Université de Lausanne, Le Génopode, CH-1015 Lausanne, Switzerland

**Keywords:** estrogen receptors, G protein-coupled estrogen receptor, nuclear receptors, vitellogenin, sexual dimorphism, liver diseases, oogenesis, yolk, growth hormone, xenoestrogens

## Abstract

The liver is a vital organ that sustains multiple functions beneficial for the whole organism. It is sexually dimorphic, presenting sex-biased gene expression with implications for the phenotypic differences between males and females. Estrogens are involved in this sex dimorphism and their actions in the liver of several reptiles, fishes, amphibians, and birds are discussed. The liver participates in reproduction by producing vitellogenins (yolk proteins) and eggshell proteins under the control of estrogens that act via two types of receptors active either mainly in the cell nucleus (ESR) or the cell membrane (*GPER1*). Estrogens also control hepatic lipid and lipoprotein metabolisms, with a triglyceride carrier role for VLDL from the liver to the ovaries during oogenesis. Moreover, the activation of the vitellogenin genes is used as a robust biomarker for exposure to xenoestrogens. In the context of liver diseases, high plasma estrogen levels are observed in fatty liver hemorrhagic syndrome (FLHS) in chicken implicating estrogens in the disease progression. Fishes are also used to investigate liver diseases, including models generated by mutation and transgenesis. In conclusion, studies on the roles of estrogens in the non-mammalian oviparous vertebrate liver have contributed enormously to unveil hormone-dependent physiological and physiopathological processes.

## 1. Introduction

The liver is a large internal organ that is essential for many important functions of the body. For example, it fulfills key roles in the metabolism of lipids, glucose, and amino acids and produces various secreted proteins and several hepatokines with regulatory functions. It also stores vitamins and secretes bile that helps the intestinal break-down of fats into fatty acids, which can then be taken into the body. In fact, one of the key roles of the liver is to filter the blood carrying nutrients coming from the digestive tract via the portal vein before it reaches the rest of the body. In addition to extracting energy, the liver also metabolizes drugs and detoxifies chemicals making them less harmful for the body by removing them from circulation. Similar to other organs, many types of diseases can develop in the liver, specifically nonalcoholic fatty liver (NAFLD/NASH), cirrhosis, autoimmune hepatitis, hepatitis A, B and C, and biliary atresia, to name but a few [1]. In contrast to mammals, in which excess accumulation of fat in the liver is detrimental, physiological accumulation of lipids in the liver has been observed in oviparous vertebrates, which are used as fuel for energy in the adverse season (fishes), during migration (birds), or as a primitive counterpart of mammalian white adipose tissue in tadpoles (amphibians) [2,3,4].

The liver parenchymal cells, called hepatocytes, alone or together with the four main types of hepatic nonparenchymal cells (Kupffer cells, hepatic stellate cells, sinusoidal endothelial cells, immune cells) sustain a very large amount of physiological and metabolic pathways as mentioned above, which are necessary for homeostasis maintenance. These roles are supported by the “metabolic zonation” of the liver parenchyma. In fact, all major metabolic pathways appear to be zonated [5,6,7]. Zone 1 or periportal zone, the best blood perfused area, localizes around the portal triads and mostly harbors oxidative metabolisms such as fatty acid β-oxidation, amino acid catabolism, cholesterol synthesis, and gluconeogenesis. Zone 3 or perivenous zone is around the central vein; it has the lowest perfusion and plays a large role in ketogenesis, glycolysis, lipogenesis, glutamine and glycogen synthesis, cytochrome P-450-based drug detoxification, and bile acid formation [8]. The midzonal layer or Zone 2 occupies the intermediary spaces in-between Zones 1 and 3. This midmodular zone is the major provider of new hepatocytes during liver homeostasis, regeneration, and regrowth [9,10].

Not surprisingly, if considering the above-listed functions, the liver plays a rather significant role in the organ-organ crosstalk that warrants the good functioning of the whole body. Moreover, the liver interacts with the endocrine and nervous systems, thereby aiding in the coordination of physiological and metabolic functions of the whole organism. The global integrative regulatory role of the liver is furthermore supported by being a storage location for fat-soluble vitamins, copper, and iron, participating in sex hormone metabolism, producing plasma carrier proteins for factors implicated in development, and clotting factors with their specific roles in hematology. Other links to hematology are roles in heme and iron metabolisms, while some of the nonparenchymal cells (Kupffer cells, Pit sinusoidal cells, stellate cells) participate in the body’s immune system and tissue repair.

Most if not all hepatic metabolic and signaling pathways are regulated at the gene transcription level, though not exclusively. Among the regulatory factors of gene activity, the members of the nuclear receptor superfamily are heavily engaged [11]. These nuclear receptors (NRs) are activated by lipophilic compounds, including hormones, which can diffuse from producing cells into target cells, including liver cells, close to or at a distance of the hormone production sites in the body. These activators comprise steroids, retinoids, thyroid hormones, vitamin D3, fatty acids, and fatty acid derivatives, all known as potent regulators of cell differentiation, development, and organ physiology [12]. Notably, the activity of these receptors can also be modulated by posttranslational modifications, including phosphorylation, acetylation, SUMOylation, and ubiquitination [13,14]. In vertebrates, the NR superfamily counts 73 members in zebrafish, 47 in rats, 49 in mice, and 48 in humans [15]. A key property of these receptors is their ability to directly interact with the control region of the genes they transcriptionally regulate while some of them also possess a so-called non-genomic effects [16]. Their target genes have key roles in both embryonic development and adult homeostasis, and they are collectively implicated in a vast majority, if not all, of the biological processes of the body [11,17]. The hepatic roles of several of these receptors have been reviewed recently and, therefore, will not be discussed herein [18,19,20,21,22,23,24,25].

Oviparous vertebrates reproduce by laying eggs generally with no embryonic development in the mother, except for ovoviviparous animals, whose embryos develop inside eggs that remain in the mother until hatching, as in some reptiles [26]. Oviparous reproduction is found in fishes, amphibians, most reptiles, birds, and monotremes (not discussed in the review). In this reproduction strategy, a fine-tuned crosstalk between the ovary and the liver is of paramount importance for the production of eggs, which in most land-living animals are protected by hard shells. Estrogens play a key role in this crosstalk, which in addition to their impacts on lipids and glucose metabolisms, control the production of the yolk and egg shell proteins [27,28,29]. After secretion, vitellogenin and VLDLs carrying lipids circulate from the liver to the maturing oocytes to build up the yolk reserve to be used as nutrient and energy sources by the embryos during development until hatching (Figure 1). Similarly, the secreted egg envelop proteins are transported in the bloodstream to the ovary. In fact, reproductive functions and liver metabolism are intimately linked in oviparous vertebrates, making the liver the most sexual dimorphic metabolic organ. Moreover, energy metabolism in the liver depends on the cross-talk of several NRs, including in addition to the estrogen receptor, peroxisome proliferator-activated receptors (PPARs), glucocorticoid receptor (GR), thyroid hormone receptor (TR), and others [11,30]. This review aims at recapitulating the multifaceted aspects of non-mammalian oviparous vertebrate estrogen-controlled liver physiology and physiopathology.

## 2. Liver Sexual Dimorphism and Hepatic Sexually Dimorphic Gene Expression

As mentioned above, the liver has a key role in coordinating a multitude of physiological processes. Extensive sexual dimorphism emphasizes the different requirements between sexes for reproduction [31], and the liver of most oviparous animals is the main production site of major oocyte constituents, such as the glycophospholipoprotein vitellogenins and the zona pellucida proteins (choriogenins), as will be discussed in detail below [32]. As a whole and just mentioned, sexual dimorphism has evolved to best serve the needs of both sexes. NRs and hormones play a critical role in regulating male- and female-specific metabolic pathways [33]. In fact, it is thought that females were especially subjected to strong evolutionary pressure to couple energy metabolism and reproduction [34]. In mammals, especially in rats, studies have revealed most of the mechanism that links both processes. A signaling cascade commencing in brain centers regulates the expression and release of growth hormone-releasing hormone (GHRH) and somatostatin to control the secretion pattern of growth hormone (GH) in the anterior pituitary gland [35,36]. The pulsatile pattern of GH secretion presents some sex differences in the amplitude and frequency of pulses [37]. In mice, the default pattern of GH secretion is feminine; it is somatostatin that masculinizes the hypothalamic–pituitary–liver axis pulses [38]. GH establishes sexual dimorphic gene expression in the liver via the expression and activation of GH receptors. The level of expression of more than 1000 genes is different in females compared to males, and close to 90% of these liver genes appear to be dependent on the sexually dimorphic GH secretion in the pituitary gland [35]. Moreover, the sex-specific release of GH in the adult is dependent on testosterone exposure during neonatal life and is then brought about by the re-initiation of testosterone production in the testis at puberty [37,39,40]. It remains to be determined to what extent the maturation, organization, regulation, and sexual dimorphism of the oviparous vertebrate hypothalamic-pituitary-gonadal (HPG) axis is identical to that of rodents, but the main features are conserved (Figure 1).

To characterize the liver sexual dimorphism in oviparous vertebrates, male–female comparative analyses of gene expression have been performed in some fishes, amphibians, and birds. These studies have focused on both specific genes and whole transcriptome expression. Small fishes, such as the zebrafish (*Danio rerio*) and the Japanese medaka (*Oryzias latipes*) have become much used oviparous vertebrate model organisms. They are suitable for studying metabolic and physiological processes and exploring the mechanisms of xenobiotic actions, including pollutants. Furthermore, the zebrafish is an excellent model to study development processes because its embryos are transparent, and it has a high reproduction rate, with close metabolic and physiological resemblance to other vertebrates [41]. Medaka models are produced from inbred lines and are characterized by high fecundity, rapid development and growth, and metabolic profiles of lipids and sugars which are close to mammals [42].

Growth analysis in mammals indicated that male mice grow faster than female mice and that the signal transducer and activator of transcription *Stat5b* is required for sexual size dimorphism and male-biased liver gene expression in mouse. In rodents, pulsatile but not continuous GH exposure activates hepatic *Stat5b* by tyrosine phosphorylation, triggering dimerization and then nuclear translocation. In the nucleus, it activates the transcription of target genes involved in regulating the sexual dimorphism of liver gene expression under the control of pulsatile serum GH levels. In support of this mechanism, the disruption of *Stat5b* results in GH pulse insensitivity leading to loss of male-characteristic body growth rates and male-specific liver gene expression [43]. Sexual body size dimorphism also occurs in many fish species. For instance, there is a significant body size difference between males and females in the adult zebrafish, which is not observed in *stat5b*-mutated fishes [44]. Interestingly, comparative whole transcriptome analysis in liver showed that the number of female-biased and male-biased genes was much lower in *stat5b* mutant zebrafish than in wild-type animals [45]. Several metabolic pathways were affected upon *stat5b* mutation, and the expression of candidate genes related to the growth traits and *stat5b* function, such as *greb1* (growth regulating estrogen receptor binding 1—an estrogen receptor-regulated tumor promoter), *lepr* (leptin receptor—involved in the regulation of body weight), and *igf2b* (insulin-like growth factor 2b—diverse functions in organ development) was modified. These observations contribute to a better understanding of the molecular mechanisms underlying sexual size dimorphism in fish [45]. Furthermore, *stat5b* is related to body weight in chicken [43,46]. In line with the above, hepatic expression of growth hormone receptors (ghr1, ghr2) and insulin-like growth factors (igf-1, igf-2), as well as plasma Igf-1 levels, were found higher in tilapia (*Oreochromis mossambicus*) males than in females, which reflects the greater male growth rate [47]. Sexual dimorphism in the liver of the adult medaka fish has been studied by an integrative approach combining metabolomics, proteomics, transcriptomics, and histology, which globally depicted a prominent sexual dimorphism of various metabolic functions of hepatocytes, including steroidogenesis and detoxification [48]. Another study investigated sex-specific co-expression networks and sex-biased gene expression in the salmonid Brook Charr *Salvelinus fontinalis,* a primarily freshwater species indigenous to Eastern North America [49]. Both co-expression gene networks and sex-biased gene expression were characterized in female and male livers during the reproductive season following an acute handling stressor. Gene networks presented a moderate to high conservation between sexes. No sex-specific module, which can be defined as a group of genes with similar expression profiles and may be co-regulated and/or are functionally related [50], was identified in females, but two were found in males, which may contribute to sexual dimorphism. In addition to these network results, differentially expressed individual gene analysis between the sexes showed that 16% of transcripts are sex-biased. Of note, these sex-biased genes and the sex-specific modules in males were not over-represented on the putative sex chromosome [49].

Similarly, there is relatively little known on the differentiation in global gene expression between females and males in amphibians. However, gene expression analysis in the liver of the Asiatic toad *Bufo gargarizans* using RNA-sequencing was performed. There were 695 differentially expressed genes between females and males, and of these, 655 genes were significantly down-regulated in males. Among these, the gene encoding NADPH-cytochrome P450 reductase isoform X1 and several genes for key rate-limited regulated enzymes of glycolysis were down-regulated in males as compared to females [51]. By contrast to most species, *Bufo gargarizans* females are larger than males, which results in a higher basal metabolic rate that increases with body surface area and mass. During hibernation, *Bufo gargarizans* females are torpid; it is when they carry developing eggs. Therefore, the females cannot reduce their metabolism as much as males to warranty normal egg development. Nonetheless, a total of 1399 genes were differentially expressed between active and torpid females. Among them, 395 genes, including genes involved in stress response and contractile proteins, had an increased expression in torpid females, which may suggest roles in protein homeostasis to preserve tissue-specific tasks and heart functions, respectively [51]. Among the 1004 genes that were down-regulated in torpid females, most concerned shifts in energy utilization and reduced metabolic rate. Between active and torpid males, there were fewer differentially expressed genes (715 genes), among which 337 were up-regulated and 378 down-regulated [51]. Down-regulated genes comprised genes involved in amino acid metabolic processes and facilitated glucose transport. Like in females, the expression of several heat shock protein (*HSP*) genes was enhanced during torpor. Interestingly, an up-regulation of *HSP* genes has also been observed in hibernating mammals [51,52].

Of note, transcriptomic data from the liver of the brown anole lizards (*Anolis sagrei*) suggest sex-specific modifiers such as testosterone induce sex-biased gene expression to support the phenotypic development of sexual dimorphism [53].

Collectively, these observations underscore how much liver sexual dimorphism in oviparous vertebrates depends on sex-biased gene expression with important implications for phenotypic variation between males and females. Nevertheless, there are still important gaps in our understanding of the relationship between transcriptional and phenotypic dimorphism [54].

## 3. Crosstalk between Liver and Ovary in the Production of Egg Yolk and Eggshell Proteins

Egg yolk, in oviparous vertebrates, contains lipids and proteins, which are nutrients that are needed for embryonic development. Parts of these nutrients are derived from proteins called vitellogenins that are produced in the liver. Vitellogenins are proteins that transport some lipids, phosphorous, and calcium, which are delivered to the growing oocytes in the ovaries and are vital for embryonic development. This essential resource cannot be provided elsewhere in the exterior to the maturing eggs [27]. In the oocytes, vitellogenin is cleaved to lipovitellin 1, lipovitellin 2, and phosvitin. In *Xenopus laevis*, there are also two smaller phosphoproteins, the phosvettes I and II resulting from an additional cleaving in the phosvitin domain of one of the vitellogenin variants [28]. The arrangement and functional implication of phosvitin and the other subdomains in the multidomain vitellogenin precursor, with respect to characteristics like lipids loading, the solubility of the yolk complexes in the circulating plasma, and other functions, have been discussed elsewhere [55]. Vitellogenin is encoded in a small VIT gene family. The genes appear to act in a dose-dependent manner since a positive correlation is observed between the gene copy number and the yolking capacity. In accordance with this proposition, fishes with multiple gene copies, produce large amounts of eggs in a shorter time than reptiles and birds [56]. This idea is also applying to frogs. For instance, *Xenopus laevis*, which has a larger body size and lays larger eggs than *X. tropicalis*, retained an additional VIT gene copy which results from increased polyploidization that is thought to contribute to meet enhanced yolk requirements [57,58].

The eggs are surrounded by a thin membrane providing an extracellular coat called vitelline envelope (VE) in oviparous vertebrates and zona pellucida (ZP) in mammals. It is composed of long, interconnected filaments made up of conserved proteins. There are three VE proteins in the rainbow trout (*Oncorhynchus mykiss*) VEα, VEß and VEγ. In the mouse (*Mus musculus*), the ZP also comprises three proteins, ZP1, ZP2, and ZP3, with ~25% sequence identity between the trout VE and mouse ZP proteins [29]. It was proposed that the genes encoding these proteins are derived from a common ancestral gene, which has been duplicated to give rise to three to four genes in fish, five genes in amphibians, six genes in birds, and three to four genes in mammals [59]. Mammalian ZP proteins and amphibian VE proteins are synthesized in the ovary, whereas fish and bird VE proteins are synthesized in either the liver or ovary or in both. It has been suggested that the expansion of gene copy number and acquisition of dual sites of synthesis (liver and/or ovary) of the egg envelope proteins originated from an ancient polyploidization event and subsequent species-specific gene duplications [29]. The secreted VE proteins are transported in the bloodstream from the liver to the ovary, where they are incorporated into the VE of the maturating oocytes (Figure 1) [29].

In the liver, the production of vitellogenins is under the strict control of estrogens. For instance, in the *Xenopus laevis* male liver, the vitellogenin genes are silent, but are strongly responsive to 17β-estradiol (E2) treatment, which promoted them as model genes to study estrogen responsiveness at the molecular level [60,61,62,63]. Besides vitellogenins, the synthesis of VE proteins in the liver is under estrogen control, too. Thus, the hormone-regulated production of yolk proteins and VE proteins in the liver depends on an important fine-tuned crosstalk with the ovary that produces estrogens and the liver that synthesizes the discussed key components of the maturing eggs (Figure 1). As observed in male tilapia, E2 treatment stimulates vitellogenin production primarily via the estrogen receptor α (*esr1*) and, importantly, down-regulates the GH/IGF-I axis, resulting in a shift in energy flux from somatic growth towards yolk production [47]. Similarly, it has been suggested that expression of *esr1* is necessary for vitellogenin synthesis in the liver of lizard *Podarcis sicula* [64]. Thus, in oviparous vertebrates, the ovary–liver crosstalk is based on E2 signaling via the estrogen receptors expressed in the liver as discussed in the next chapter that also describes the occurrence of estrogen receptors in the different oviparous species.

## 4. Estrogen Receptors in Oviparous Vertebrates and Their Hepatic Expression

The evolution of steroid receptors in oviparous vertebrates conferred a crucial advantage to this lineage because they have become master regulators of vital functions such as reproduction, development, cell and organ differentiation, and immune responses. Therefore, they have contributed to the evolutionary success of all vertebrates. The estrogen receptor first appeared in cephalochordates (amphioxus), the close ancestors of vertebrates (Figure 2). Different nomenclatures for estrogen receptors have been used in the literature, which created some confusion. In mammals, the estrogen receptor-α subtype is designated *ESR1* and the β subtype *ESR2* (see HUGO Gene Nomenclature Committee; https://www.genenames.org/tools/search/ (accessed on 6 July 2021)). Below, we will use this nomenclature for mammals and, for non-mammalian vertebrates, we are using the now well-accepted zebrafish nomenclature (see Official Zebrafish Nomenclature Guidelines; https://zfin.org/ (accessed on 6 July 2021)). According to these guidelines, the ERα subtype is designated *esr1*, the ERβ1 subtype *esr2b*, and the ERβ2 subtype *esr2a*. Estrogen receptor subtypes emerged in cartilaginous fishes (Chondrichthyes like sharks), which have skeletons composed of cartilage rather than bone. This duplication of the receptors promoted the diversification of signaling by estrogens that readily diffuse through cell membranes [65,66,67]. As mentioned above for zebrafish, the teleost lineage has at least three estrogen receptors, *esr1* (α subtype), *esr2b*, and *esr2a* (β subtypes), which are the results of a whole genome duplication that occurred in the evolution of this lineage [68,69,70]. Interestingly, estrogen receptors in oviparous vertebrates bind E2 with an affinity similar to that reported in mammals [71,72]. Some fish species, such as the rainbow trout, the Atlantic salmon, and the Pengze crucian carp, even present four isotypes, two α and two β isotypes [66,69]. Finally, in amphibians, two (*esr1*, *esr2*) or three (two *esr1s*, *esr2*) isotypes of estrogen receptor have been identified in *Xenopus laevis* and *Xenopus tropicalis*, respectively [71,73]. Compared to the estrogen receptors in mammals, bird *esrs* are the most highly conserved [66]. The occurrence of *ESRS* in oviparous vertebrates is summarized in Figure 2.

In the liver of the rainbow trout, one of the characteristics of *esr1* is the huge upregulation triggered by its ligand, which is relevant for the strong induction of vitellogenin by estrogenic stimulation [75]. In fact, the regulation of esr expression in the oviparous vertebrate liver received much attention in the context of vitellogenesis. The expression of hepatic *esr1* is stimulated by estrogens and correlates well with the upregulation of vitellogenin production. A recent study on the hepatic expression profiles of both vitellogenin and estrogen receptor subtypes during vitellogenesis in the female yellowtail fish suggests that the three yellowtail vitellogenin subtypes are regulated by the three esr subtypes jointly, with *esr1* playing a key role [76]. An estrogen-responsive elements (ERE) was found in the promoter region of the rainbow trout *esr1* gene [77]. In the zebrafish *esr1* promoter, there is an imperfect ERE, together with an AP-1/ERE half-site [78]. Furthermore, E2 stabilizes the *esr1* mRNA in the liver of the rainbow trout, which is an additional mean to increase *Esr1* expression by a mechanism involving an RNA-binding protein mediating the E2-associated stabilization process that appears to be conserved across species [79]. ERE half-sites associated with SP1 sites have also been identified in the promoter of the *esrb2* (coding for ERβ1) promoter of zebrafish [80]. The autoregulation of the *esrs* in the liver of several fish species has been reviewed previously [81], as was the work on goldfish and other fish species, which has shown that hepatic *esrs* are under homologous estrogen regulation in a dose-, subtype-, time-, and sex-specific manner. Furthermore, these changes in regulation depend on the reproductive stage of the fish [81,82,83,84]. These findings collectively underscore the autoregulation of the esr by E2 and the putative functional interaction of the esr with other transcription factors such as SP1 and AP-1 (see below).

In mammals and non-mammalian vertebrates, *ESRs*/*esrs* form dimers to regulate estrogen-responsive target genes. Estrogen receptor isotypes can form homodimers as well as heterodimers with each other. When activated by estrogens, they interact with coregulator complexes for the stimulation of responsive genes through binding to one or more palindromic ERE in the regulatory region of these target genes [85] (Figure 3A). The nucleotide sequence of the consensus ERE is GGTCAnnnTGAC, although perfect elements are relatively rare. The ERE was originally identified in the promoters of the vitellogenin genes, which were the first well-characterized estrogen-responsive genes and played a key role in deciphering the molecular mechanistic action of the *esrs* [86,87,88]. At the gene promoters, functional interactions between the *esrs* and other transcription factors can occur, like with Sp1 in the promoter of the *Xenopus laevis* vitellogenin gene A1 [89], CTF-1, HNF3, and C/EBP in the vitellogenin B1 gene promoter [90,91,92] (Figure 3A).

Furthermore, the oviparous vertebrates and mammals have one copy of the G protein-coupled estrogen receptor (*GPER1*; *gper* in non-mammalian vertebrates) [66,93] with, so far, the exception of the European eel and the European sea bass, which have two genes coding for *gper*, *gpera* and *gperb* [74,94] (Figure 2). Both the affinity for estrogens, evaluated by E2 binding, and the signaling mechanism of this membrane *GPER1*/*gper1* are well conserved between these distantly related vertebrates [69,95]. As seen above, the genomic regulatory mechanism consists of E2 binding the intracellular *esr1* and *esr2* to activate them to regulate gene transcription directly via EREs over hours or days. However, the non-genomic E2 signaling via *gper* pathway is different. The hormone binds to the cell membrane *gper1* to activate second messengers comprising signaling via kinases or calcium flux with outputs within seconds or minutes [96] (Figure 3B). The study of the functions of *gper1* has so far concentrated on sexual dimorphism in immune, cardiovascular, renal, and brain functions. Interestingly, environmental chemicals with estrogenic characteristics can also exert their biological effects through GPER-mediated signaling pathways [97].

Moreover, little is known as discussed in the next chapter on its role in the oviparous vertebrate liver [69,98,99,100]. Further explorations of *gper1* functions both independently and as a cross-talk partner of the classic *esrs* in both genders promise to hold precious information for the understanding and possible therapeutic potential of this membrane receptor.

In conclusion, the expression of the classic estrogen receptor and *gper1* in the liver of female oviparous vertebrates is key for producing supplies (yolk and VE proteins) to maturing eggs in the ovaries and requires a marked degree of hepatic sexual dimorphism. Furthermore, the hepatic receptors for estrogens and androgens (not discussed herein) show a marked degree of sexual dimorphism and are expressed at a significant level when the need for the hormone action is most critical [31].

## 5. Estrogen-Dependent Lipid and Lipoprotein Metabolisms in the Oviparous Liver

As mentioned earlier in the review, the liver is an essential organ for regulating energy homeostasis, and NRs play important roles in this process [11]. In mammals, the orphan estrogen-related receptor (ERR) subfamily, which comprises three isotypes ERRα, ERRβ, and ERRγ, is a central regulator of energy metabolism. ERRα, in particular, has been identified as regulating lipid and carbohydrate metabolism and mitochondrial activity and was proposed as a potential target for the treatment of hepatic metabolic disorders and associated diseases [101]. Compared to mammals, relatively little is known on the roles of ERRs in the hepatic energy homeostasis of non-mammalian oviparous vertebrates. In zebrafish, five errs were identified, and the homolog of the mammalian ERRα was not found, so far, to be associated with energy metabolism, but identified as a transcriptional regulator of morphogenetic movements during gastrulation [102,103]. Four err genes were found in the killifish *Fundulus heteroclitus,* indicating that fishes present a higher diversity of errs relative to mammals. Furthermore, their expression appears also to be regulated differently relative to mammals [104]. Obviously, further work is needed to determine conserved and divergent ERR gene functions between oviparous vertebrates and mammals.

As already reviewed earlier for the female mammalian liver, estrogens regulate metabolism via its receptors by enhancing cholesterol secretion, glucose catabolism, and lipolysis, while decreasing fatty acid uptake, lipogenesis, and gluconeogenesis [105]. When compared to mammals, oviparous vertebrates present several differences in the metabolism of lipids, including transport of dietary lipids to the liver, hepatic lipogenesis, unique lipoproteins in the blood [106]. A strong and sophisticated coordination, under hormonal control, is needed in the female liver between protein synthesis (vitellogenin, egg shell proteins, lipoproteins) and lipid metabolism. It allows a synchronized transport of yolk components from the liver to the ovary during the egg-laying period, which determines the success of reproduction [107,108]. Transcriptome profiling of liver samples revealed 1046 differentially expressed transcripts during vitellogenesis in zebrafish. More than 60% of these genes were regulated by E2, including 15 genes in the Gene Ontology (GO) function “lipid metabolic process”, which supports a close association between vitellogenesis and lipid metabolism [109]. Similarly, transcriptome profiling of liver in egg-laying hens suggests that the majority of changes at the gene expression level are associated with fat metabolism [110]. Studies in estrogenized chicken demonstrated hormonal stimulation of fatty acid synthesis through upregulation of the activity of acetylCoA carboxylase and fatty acid synthase together with enhanced incorporation of fatty acids in triglycerides and phospholipids. Interestingly, under the influence of estrogens there are relatively more monounsaturated and less saturated and polyunsaturated fatty acids in the liver, whereas hepatic Δ9 desaturase activity is enhanced. Furthermore, there is an association of fatty acid synthesis and desaturation with an increased VLDL secretion, which is beneficial because it limits the level of hepatic triglyceride accumulation [111]. In egg-laying chicken, the production of triacylglycerols (TG), free cholesterol, cholesteryl esters, phospholipids, free fatty acids, and proteins is actively increased in the liver. This enhancement correlates with the estrogen levels that rise significantly during the egg-laying period. As observed in the chicken liver, the glycerophosphate synthesis pathway is key for TG production. Several genes in this pathway are regulated by estrogens. *Gpam* (glycerol-3-phosphate acyltransferase 1) and *agpat2* (1-acyl-sn-glycerol-3-phosphate acyltransferase beta) are upregulated while *agpat3* (1-acyl-sn-glycerol-3-phosphate acyltransferase gamma), *agpat9* (1-acylglycerol-3-phosphate O-acyltransferase 9), *lpin1* and *lpin2* (phosphatidate phosphatase LPIN1 and LPIN2) are down-regulated by E2. Further studies are needed to clarify the precise role of these genes in avian hepatic lipid metabolism [112]. The expression of the key lipogenic factor thyroid hormone-responsive spot 14 (thrsp 14), an important transcription factor in hepatocytes, which responds rapidly to thyroid hormone and controls the expression of several lipogenic genes, increases with female sex maturation reaching its highest levels at the peak of egg production in chicken. The *thrsp* gene was recently identified as a direct estrogen receptor target gene belonging to the liver estrogen regulation network [113]. Long noncoding RNAs (lncRNAs) are also involved in hepatic lipid metabolism. A study compared the difference in lncRNAs expression in the livers of pre-laying and peak-laying hens and investigated the interaction networks among lncRNAs, mRNAs, and miRNAs [114]. A series of lncRNAs associated with lipid metabolism were identified by transcriptome sequencing and functional analysis, and the study found that some of them might be regulated by both the lncRNAs and miRNAs. One of the upregulated lncRNAs, named lncLTR (a lncRNA acting as a liver triglyceride synthesis regulator) is induced by estrogens via *esr2* and is associated with chicken carcass trait and blood triglyceride content [114]. The peptide Apela, a ligand for the G-protein-coupled apelin receptor, was first identified in embryonic stem cells. The chicken Apela homologue was characterized recently and found to be highly expressed in liver at the peak-laying stage. Furthermore, it was also significantly up-regulated in the liver by E2 treatment and proposed to be involved in hepatic lipid metabolism, but this role remains unclear [115]. In frogs, there are also plasma estrogen fluctuations that are circannual with the highest level during the egg production period in females. These fluctuations impact both cholesterol metabolism as they play a major role in the metabolism of isoprenoid compounds by mechanisms implicating the expression of 3-hydroxy-3-methylglutaryl coenzyme A (HMGCoA) reductase and that of low-density lipoprotein receptor (LDLr) as observed in the liver of *Rana esculenta*. However, the expressions of LDLr and HMGCoA reductase are regulated by different mechanisms after 17α-ethynyl-estradiol stimulation. An increase in HMGCoA reductase is controlled at the post-transcriptional level, while that of the LDLr is regulated transcriptionally [116]. Furthermore, vitellogenin secretion is promoted by microsomal triglyceride transfer protein (MTP) [117]. However, up-regulation of MTP in the chicken liver is not necessary for the increased VLDL assembly during egg production, suggesting that MTP is not rate-limiting for the massive estrogen-induced secretion of VLDL during the egg-laying cycle [118].

Gallinaceous avian species have been selected for the highest possible egg production. In a 60 g hen egg, the yolk contains 6 g of TGs that were transported to the oocyte in the ovary from the liver during the laying period in apolipoprotein B 100—containing particles. During this period, estrogens shift the lipoprotein production in hepatocytes from so-called generic VLDL to yolk targeted VLDL (VLDLy). The VLDLy particles are smaller (~30 nm) than generic VLDL particles (~70 nm) and are coated with apolipoproteinVLDL-II that confers resistance to lipoprotein lipase hydrolysis (Figure 4). Whereas the function of generic VLDL is to provide TGs to the whole body for utilization in the different tissues or storage in adipose tissue for later use, the role of VLDLy is to specifically carry TGs to the oocytes, where they will serve as an energy source for the developing chicken. After reaching the maturing oocytes through the ovarian capillaries, they are taken up by receptor-mediated endocytosis and provide the caloric requirements of the future developing chick [119]. The assembly of VLDLy particles in the liver most likely proceeds according to a two-step process of apolipoprotein B100 core lipidation. The first step, in which MTP is involved, is the formation of a VLDL precursor that proceeds through a second step to form VLDL particles [120]. It was suggested that this two-step mechanism of apoB core lipidation, also found in mammals, is an ancient mechanism that appeared in oviparous vertebrates [119]. It can be speculated that non-gallinaceous avian species that have not been selected for maximum egg production would exhibit less dramatic estrogen-induced shifts in lipid metabolism during egg production. This hypothesis was tested in a passerine bird, the zebra finch. Interestingly, zebra finches and chickens showed opposing shifts in the distribution of VLDL particle size during egg production. However, this change still preserves an important production and maintenance of a large proportion of small VLDLy particles to be incorporated into newly forming egg yolk [121]. Interestingly, DNA sequence analyses suggest that reptiles may express an Apo VLDL-II-like protein targeting triglycerides to yolk via VLDLy as seen in birds [108].

## 6. Hepatic Crosstalk between *ESRS* and Other Members of the Nuclear Receptor Superfamily

Given the paramount role of the liver under the control of estrogens in reproductive physiology, it is interesting to look for functional interactions between the estrogen receptor and other members of the nuclear receptor superfamily. Indeed, the oviparous vertebrate liver is a hub for physiological control by nuclear receptors. First of all, esr and *gper* functionally interact in the regulation of vitellogenesis in zebrafish. *Gper* promotes vitellogenesis directly via its nongenomic action, whose precise mechanism is not yet elucidated, and also by interaction with the nuclear *esrs*. Activation of *gper* stimulates the expression of *esr1* (ERα) and *esr2a* (ERβ2), but down-regulates *esr2b* (ERβ1). Activated *esr2b* and *esr2a* increase the expression of *esr1* and, in turn, *esr1* enhances *gper* gene activity [122] (Figure 3C).

After the characterization of the ERE as mediator of the stimulatory effect of esr on the expression of vitellogenin genes, further studies assessed whether other nuclear receptors could interfere with esr signaling through this response element. In gene transfection experiments using the *Xenopus* VIT A2 ERE in a reporter gene, the retinoid X receptor β (RXRβ) did inhibit esr-dependent stimulation via the ERE by two mechanisms. The first one involved RXRβ and an unidentified nuclear factor; the second one was thyroid hormone-dependent via the RXRβ-thyroid hormone receptor α heterodimer [123] (Figure 5A). A similar approach was used to examine a possible implication of RXR and its ligand 9-cis-retinoic acid (9-cis-RA) in regulation of the ERE-containing apoVLDLII promoter. Further, 9-cis-RA down-regulated the estrogen-induced expression of a transfected estrogen-responsive VLDL reporter gene through the main ERE in the gene promoter. In chicken hepatoma cells, 9-cis-RA also had a strong repressive effect on the estrogen-induced expression of the endogenous apoVLDLII gene [124] (Figure 5A). Again, using transgene transfection, it was shown that the RXRβ-PPARα heterodimer can activate an estrogen-responsive reporter gene when activated by 9-cis-RA, an effect mediated by the binding of RXRβ-PPARα to the ERE [125] (Figure 5).

This observation suggested that 9-cis-RA can activate the RXRβ-PPARα heterodimer, which opens interesting perspectives for a more complex regulation of esr target genes. However, the natural vitellogenin A2 gene promoter in its chromosomal context revealed that the EREs could not transactivate the gene via RXR-PPAR bound to the ERE due to nonpermissive promoter configuration. Nevertheless, RXR-PPAR can inhibit transactivation by the estrogen receptor via competitive ERE binding, revealing a potential signaling cross-talk between estrogen receptor dimers and RXR-PPAR heterodimers [126]. 

Estrogens and thyroid hormones direct the initiation and progression of vitellogenesis and metamorphosis in frogs raising the question of a potential interplay between these vital processes. In fact, exogenous triiodothyronine (T3) can trigger precocious estrogen-dependent activation of the vitellogenin genes during metamorphosis in *Xenopus laevis*. T3 stimulates thyroid hormone receptor (TR) expression, allowing early activation of the silent vitellogenin genes by E2 at the metamorphic climax, but not before mid-metamorphosis. This developmental stage functional interaction between both hormones determines the onset of estrogen receptor and vitellogenin gene activation during *Xenopus* postembryonic development [127]. Similar priming with T3 of estrogen-dependent vitellogenin gene expression occurs in the liver of female goldfish (*Carassius auratus*). T3 up-regulates the expression of the *esr1* in a TRα-1 and TRβ-dependent manner. Therefore, T3 primes the liver by upregulating *esr1* to allow a strong subsequent stimulation of vitellogenin production by E2 (Figure 5B). This cross-talk supports a key physiologic mechanism in the context of high circulating levels of thyroid hormones during early gonadal recrudescence, which sustains egg development by producing large amounts of vitellogenin [128].

Only a few studies are assessing the respective roles of progestins, glucocorticoids and estrogens on vitellogenesis. First, vitellogenin production was studied during the annual reproductive cycle of the deserticole oviparous lizard, *Uromastyx acanthinura*, with respect to hepatic expression and distribution of estrogen and progesterone receptors according to the period of the reproductive cycle and the experimental administration of E2. Only the *esr2* subtype was present at variable localization during the cycle, and it was nuclear during vitellogenesis. On the contrary, the progesterone receptors were detected only in the luteal phase and during sexual rest but not during vitellogenesis. Furthermore, it was completely absent after exogenous administration of E2 to females in sexual rest. These observations suggest that in this lizard, *esr2* mediates the effect of E2 in vitellogenin synthesis and that the progesterone receptors are necessary for the repressive effect of progesterone on the hepatic vitellogenin production during sexual rest [129]. Second, the *Xenopus* VIT A2 promoter has a “simple” esr binding ERE, in contrast to the promoter of the chicken VIT II gene presenting a multihormonal response element for estrogens, progestins, and glucocorticoids. In conclusion, these *Xenopus* and chicken vitellogenin genes present different promoter configurations, the second responding to three different steroid hormones, at least in an experimental context [130]. The biological significance of this difference is unknown so far as there is no in-depth molecular study on the regulation of the VIT genes in the chicken liver. Nevertheless, one can speculate that this complex hormone response element might allow a fine-tuned regulation during changing physiological conditions.

Fish farming has suggested that stress and its induced rise in cortisol levels reduce vitellogenesis. This idea was investigated in the rainbow trout *Oncorhynchus mykiss*. It was found that cortisol implants in maturing fishes reduce hepatic esr and vitellogenin synthesis and that this effect was likely to take place at the transcriptional level. Cell culture experiments showed that the rainbow trout glucocorticoid receptor strongly inhibited the E2-stimulated transcriptional activity of the esr promoter. Together, these observations indicate that the GR mediates a transcriptional interference on esr expression which correlates with the negative impacts of stress on vitellogenesis [131] (Figure 5C). Cell culture experiments in breast cancer cells indicated that a direct interaction between GR and *esr1*, which is mediated via the transcription factor activator protein 1 (AP-1), has an important role in controlling *esr1* activity and GR-mediated inhibition of E2-induced cell proliferation [132].

These studies have collectively revealed how much the esr functions are interwoven with other members of the nuclear receptor superfamily which, in physiology and metabolism, is the rule rather than the exception.

## 7. The Oviparous Vertebrate Liver as a Sensor of Estrogen Disrupting Compounds

Endocrine-disrupting chemicals (EDCs) represent a class of chemicals that interfere with hormone signaling pathways to impair their proper function. Nuclear receptors are the major targets of these substances [133]. The liver, which also orchestrates the elimination of these chemicals, is a key target for their action and presents sexual dimorphism in coping with them [134]. In fact, the *ESRs* bind a variety of substances, ranging from natural or synthetic compounds (xenoestrogens) and many industrial EDCs. The natural endogenous estrogens (estradiol, estriol, estrone) are high-affinity ligands compared to the phytoestrogens (genistein and ferutinin), pesticides, or industrial plasticizers that bind to *esrs* with affinities in the sub-to micromolar range concentrations [133,135].

As discussed above, the vitellogenin genes are under the control of estrogens in the liver of oviparous vertebrates [27]. Thus, it was already proposed more than 25 years ago that vitellogenesis in fish could be used as a biomarker for estrogenic contamination of the aquatic environment. When male fishes are exposed to estrogenic compounds present in the water, their otherwise silent vitellogenin genes are activated and transcribed at elevated levels in the liver, which can be measured at the mRNA and protein levels [136]. Activation of these genes in several species has become and remains a gold standard robust biomarker for exposure to estrogenic compounds in the water [137,138,139,140,141,142,143,144,145]. Recently, an ultrasensitive label-free electrochemical immunosensors for detecting marine medaka (*Oryzias melastigma*) vitellogenin was developed, and its reliability for the detection of xenoestrogens in the marine environment was validated [146].

Although models for sensing xenoestrogens in the water have received much attention, terrestrial oviparous vertebrate models for sensing soil contamination by estrogenic compounds are of high importance, too. The most used experimental model for ecotoxicological analyses is the Italian wall lizard *Podarcis siculus* [147,148]. Using this model, it was shown that glyphosate causes a severe hepatic condition associated with an upregulation of *esr1* and vitellogenin gene expression [149]. Similarly, it was shown in this model that manure applied in organic farming has estrogen-like effects that increased the expression of vitellogenin and *esr1* in hepatocytes [150]. In addition to showing that the induction of vitellogenin synthesis in the liver of males of this lizard is an excellent biomarker of soil contamination by xenoestrogens, these observations also suggest that the welfare of terrestrial wild animals could be affected by this type of pollutants.

One of the most studied compounds in aquatic toxicology is the synthetic estrogen 17alpha-ethinylestradiol (EE2), a main component of the birth control pill, which affects aquatic oviparous vertebrates at very low concentrations. Its potential hazards as an environmental estrogenic substance have been addressed using omics technologies. Studies reporting transcriptome responses to EE2 in different fish species (e.g., zebrafish, rainbow trout, fathead minnows, mummichog, pipefish, stickleback, cod, and others) have been summarized recently [141]. In these works, special attention has been given to the liver in which water contaminating EE2 stimulates gene networks related to amino acid activation and protein folding while suppressing those related to xenobiotic metabolism, immune system, TG circulation, and storage. Research done so far on EE2, and other environmental estrogenic compounds, may serve as example for assessing other harmful compounds [141]. In amphibians, the species *Xenopus tropicalis* and *Xenopus laevis* are the most used in endocrine disruptor studies on development and physiology. Exposure to EE2 discussed above for teleosts also affects *Xenopus laevis* hepatic physiology. In addition to an increase in vitellogenin gene expression, EE2 also causes a decrease in the expression of hepatic heme oxygenases 1 and 2, which are responsible for the oxidative cleavage of heme groups to generate biliverdin, carbon monoxide, and release of ferrous iron, and of biliverdin reductase A, which converts biliverdin to bilirubin. These observations suggest that EE2 can impair hemoglobin catabolism in the liver [151].

The EE2 effects as a pollutant have been relatively well studied also in mammalian species. However, whether in vitro mammalian high-throughput assays accurately reflect the estrogenic chemical–esr interactions in non-mammalian species have been addressed recently, especially for *esr1*. In this study, a cross-species comparison of compound-esr1 interactions was done, which included data on esr structural features, in vitro ligand-receptor binding and gene transactivation, as well as in vivo effects on estrogen-signaling pathways. The results showed that compounds with moderate to high estrogenic activity in mammalian systems were also effective chemicals in oviparous vertebrates. However, mammalian-based assays may not well document the *esr1* interactions of low-affinity compounds in fish and reptiles [152].

In conclusion, important further work is necessary to evaluate the perturbations of esr signaling pathways in the liver and other organs caused by a plethora of estrogenic chemicals and their impact on the whole organism.

## 8. Estrogen Associated Liver Diseases in Oviparous Vertebrates

Besides their role in liver physiology, estrogens have been implicated in the development of liver diseases, including excessive fat accumulation [153]. The most studied liver disease in oviparous vertebrates is the fatty liver hemorrhagic syndrome (FLHS) in chicken. It is a common noninfectious disease characterized by disorders in the metabolism of lipids, steatosis, liver rupture, and hepatorrhagia. It causes a dramatic drop in egg production and increased death of laying hens resulting in considerable economic losses worldwide. Nutritional (high energy diets with high calories/proteins ratio, low calcium consumption), environmental (stress, old birds), genetic (high performance strains of laying hens) and hormonal factors as well as toxic substances (mycotoxins) participate in the etiology of FLHS, but some of the molecular mechanisms remain to be investigated [154,155,156,157]. However, estrogens were identified as a factor implicated in the development of FLHS already more than forty years ago [158] and high plasma estrogen levels found associated with FLHS in hens [159]. Experimentally, FLHS can be induced by estrogen injection in laying hens fed *ad libitum* with a standard commercial layer diet. The hormone treatment disturbs lipid metabolism that causes an inflammatory response, which is most likely involved in the pathogenesis of FLHS. Interestingly in this model, not all laying hens develop FLHS while all present fatty livers. The reason for this difference in disease progression between hens remains to be elucidated [160]. Interestingly, after E2 treatment, younger layers are less susceptible to FLHS than older layers, and blood levels of TG, cholesterol, and HDL-cholesterol, but not LDL-cholesterol levels, are indicators for the overall susceptibility to FLHS in older layers [161]. Another study assessed the effects of E2 administration on lipid metabolism also in hens. E2 upregulates the synthesis of fatty acids and triacylglycerols, and the accumulation of hepatic lipids correlated with higher mRNA expression of genes related to lipid metabolism, such as *pparγ*, *acly*, *fas*, and *apoB* [153]. More recently, genome-wide H3K27ac profiles were compared to transcriptomes in liver samples from FLHS and healthy chickens. A widespread H3K27ac dysregulation in the liver of FLHS-affected chickens was observed, and FLHS-associated variations in H3K27ac marks were associated with known FLHS risk genes involved in lipid and energy metabolism (*pck1*, *apoa1*, *angptl4*, and *fabp1*) and immune function (*fgf7*, *pdgfra*, and *kit*). Furthermore, the PPAR signaling pathway was most significantly enriched in FLHS [162]. Interestingly, this pathway was also highlighted in an integrated analysis of the methylome and transcriptome in FLHS chicken. This study revealed that hypo-methylated up-regulated genes were mainly enriched in lipid metabolism-related pathways, such as fatty acid biosynthesis, fatty acid metabolism, and PPAR signaling. On the contrary, hyper-methylated down-regulated genes were related to pathways in cellular junction and communication. These observations provide evidence that steatosis and hepatocyte rupture represent important pathways controlled by DNA methylation in FLHS chickens [163]. Collectively, these studies on FLHS underscore similarities between this disease in chicken and non-alcoholic fatty liver disease (NAFLD) in mammals, with respect to lipid metabolism, hormonal regulation (sexual dimorphism), and the importance of the PPAR pathways [23,164,165]. However, a major difference is that estrogens appear to promote FLHS in the chicken, while in mammals, they are counteracting the development and progression of NAFLD, which is a metabolic inflammatory-based liver disease with a clear sex-specific male predominance [165,166]. It is noteworthy that in force-fed palmipedes, such as geese, which develop hepatic steatosis due to repeated high ingestions of a highly caloric diet, the synthesis of triacylglycerols is enhanced by estrogens, but their secretion as VLDL is very efficient and prevents liver steatosis almost completely [167]. Consequently, only male ducks are force-fed for producing foie gras.

In the overfed zebrafish with steatosis model, ovarian senescence in old female fish, characterized by very low circulating E2 levels and atretic follicles in the ovaries, promotes both the development of massive hepatic steatosis and the fibrotic progression of liver disease. By contrast, young female fish are less steatotic and do not develop fibrosis. Together these observations support the hypothesis of a protective role of estrogens and the implication of ovarian senescence, causing a strong hypoestrogenemia, in the progression of steatosis to fibrosis [168,169]. In addition to zebrafish, other fish models, such as medaka and rainbow trout, will also be useful in physiopathology for studying estrogen action in the liver, which experimentally can be relatively easily achieved in zebrafish and medaka, and also allow an easier genetic approach compared to other species [170,171,172,173].

## 9. Conclusions

Estrogen signaling in the liver of oviparous vertebrates is of the highest importance for their reproduction, not least because of its role in egg production. By contrast, mammals have gone through a major phenotypic transition and have evolved new resources to feed their developing and early offspring, such as placentation and lactation [58]. Consequently, the vitellogenin genes have progressively lost their functions in the mammalian lineage, with the most recent inactivation about 30–70 million years ago [58]. Earlier findings on yolk protein production and the genes encoding the major yolk proteins, vitellogenins, and their regulation by estrogen in *Xenopus laevis* have proved useful for investigating vitellogenesis and, importantly, the molecular mechanisms of esr action as a whole [27,174,175]. The very broad range of oviparous vertebrates, including reptiles, amphibians, fishes, and birds, offers a huge opportunity to conduct more in-depth comparative analyses of the role of estrogens in liver physiology, including the molecular bases of sexual dimorphism in animals occupying all possible ecosystems and using a diversity of nutritional sources. It can be anticipated that the significant knowledge gain resulting from this research activity will impact different areas and fields, as briefly discussed below.

The prevalence of liver diseases is high, and the difference in susceptibility between sexes is still not well understood. Thus, a deep understanding of liver physiology and pathophysiology, in particular of its hormonal component at the molecular level, should help to identify biomarkers for prognostication and better sex-specific treatments in animals and humans for conditions such as metabolic syndrome, NAFLD/NASH, hepatocarcinoma, and the sexually dimorphic immune responses on pathogenesis during Hepatitis B (HBV) and Hepatitis C (HCV) viral infections in which estrogens are implicated [165,176,177,178,179,180,181].

Eggs are an excellent, sometimes called “miracle food” much appreciated in developing as well as developed countries. They have the same biological value as breast milk with close to 40 different proteins, some with bactericidal, antigenic, and antihypertensive properties. They also contain essential vitamins, are rich in minerals, and present the desirable proportion of unsaturated and saturated fatty acids but do not contain carbohydrates and transfats. However, eggs also contain 150–180 mg of cholesterol per egg, which corresponds to 50–60% of the recommended daily intake [182,183]. Therefore, the best possible knowledge of all biological processes involved in egg production, under estrogen control, is essential in association with animal welfare, animal health, and food and environmental safety and will contribute to the optimal production of this “miracle food” called to be a major component of nutrition of populations all around the globe. Especially, the benefit of eggs to maternal and child nutritional requirements is not yet fully appreciated, mainly in developing countries, where some taboos around egg consumption have still to be dismantled [184].

As it was presented herein, egg production is an extremely fine-tuned process regulated in the liver by estrogens and by several hormones in the female ovaries, often determined by the season. These yolky eggs vary enormously in size depending on the species and are laid in large numbers in water or on land and are generally fertilized externally during interactions with males, which occur in different forms [185]. It is obvious that the tightly hormone-controlled egg production is vital for the long-term survival of oviparous vertebrate species. Disturbances, often caused by external factors, such as endocrine-disrupting pollutants in water can ruin fish populations [32,186]. The very high sensitivity and responsiveness of the oviparous vertebrate liver to estrogens and xenoestrogens have promoted the development of assays using vitellogenin as an informative biomarker for endocrine-disrupting effects of chemicals and effluents. Fishes are the most used test organisms worldwide [141,187,188,189,190] to monitor water quality, contributing to human health in all continents. Interestingly, lizards are also excellent model organisms to monitor contaminating soil xenopestrogens [147,148].

As summarized in this review, investigations on the roles of estrogens in the oviparous vertebrate liver have already contributed enormously to different fields by unveiling hormone-dependent molecular mechanisms: liver metabolism and physiology in health and disease, reproduction biology, nutrition, and assessment of endocrine-disrupting chemicals in the environment (effluents, water reservoirs, rivers). However, much more is to come from a more extensive and combined application of all omics technologies for a better global understanding of the roles of liver and estrogens in whole organisms.

## Figures and Tables

**Figure 1 metabolites-11-00502-f001:**
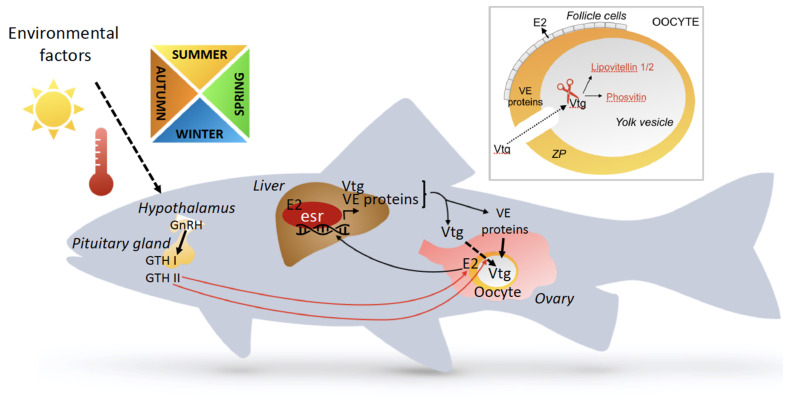
Crosstalk between liver and ovary in the production of egg yolk and eggshell proteins. Fish is taken as example in this figure. Various environmental factors stimulate the hypothalamus to secrete GnRH. In response to GnRH, the pituitary gland secretes GTHs, which stimulate both the production of 17β-estradiol (E2) in follicle cells and vitellogenin uptake in the oocytes. Vitellogenins are proteins representing the principal source of nutrients in the egg yolk of oviparous vertebrates. They are produced in the liver under the strict control of E2, secreted into the bloodstream and are taken up under the control of GTH by the growing oocytes in the ovary and cleaved into lipovitellins and phosvitins. E2: 17β-estradiol; esr: estrogen receptor; Vtg: vitellogenin; VE: vitellin envelope; GTH: gonadotropin; GnRH: gonadotropin releasing hormone. 
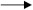
 transport; 
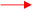
 stimulation.

**Figure 2 metabolites-11-00502-f002:**
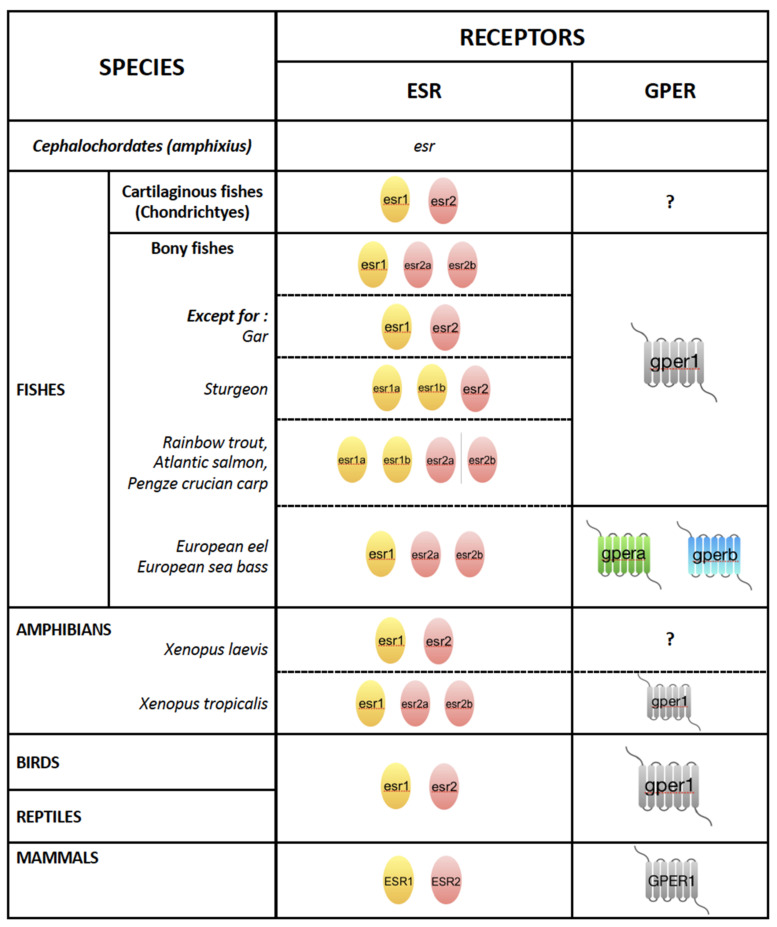
Occurrence of *ESRS* and *GPERS* in vertebrates. Detailed information can be found in [66,68,69,74].

**Figure 3 metabolites-11-00502-f003:**
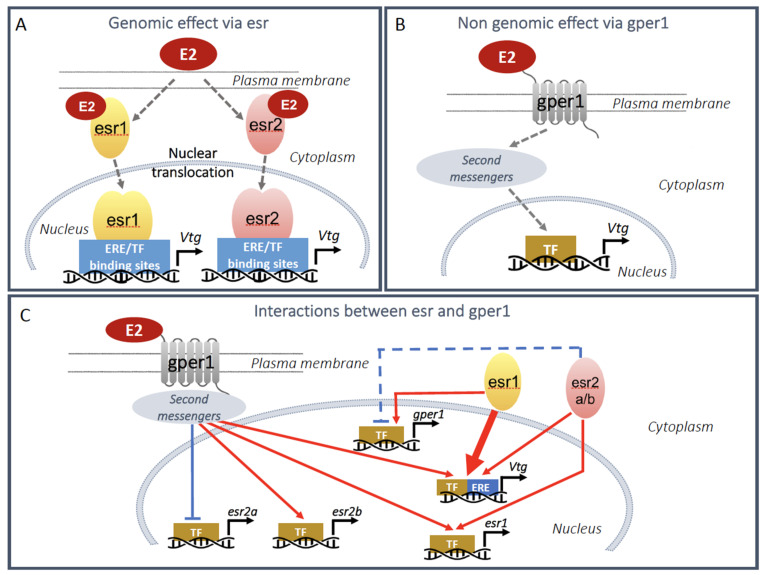
Signaling pathways of estrogens on vitellogenesis synthesis and interaction between esr and *gper* in oviparous vertebrates. (**A**) Genomic effect of E2 via *esr1* and *esr2*. When bound to the ERE in the regulatory region of the vitellogenin genes, E2-activated *esr1* or *esr2* dimers (depicted separately) stimulate the transcription of the vitellogenin genes. The ERE bound esr dimers may cooperate with other transcription factors (TF) bound to nearby TF binding sites (see text). (**B**) Non-genomic effect of E2 via *gper1*. E2-activated *gper1* stimulates a cascade of second messengers resulting in vitellogenin gene stimulation by transcription factors (TF). (**C**) Functional interactions between esr and *gper1*. *Esrs* and *gper1* cooperate in the regulation of the esr, *gper*, and vitellogenin genes. E2: 17β-estradiol; Vtg: vitellogenin; esr: estrogen receptor; TF: transcription factor (Sp1, CTF-1, HNF3, C/EBP and others); ERE: estrogen responsive element; *gper1*: g-protein coupled estrogen receptor 1. 
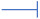
 inhibition; 
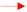
 stimulation. The blue dashed line indicates an effect to be confirmed.

**Figure 4 metabolites-11-00502-f004:**
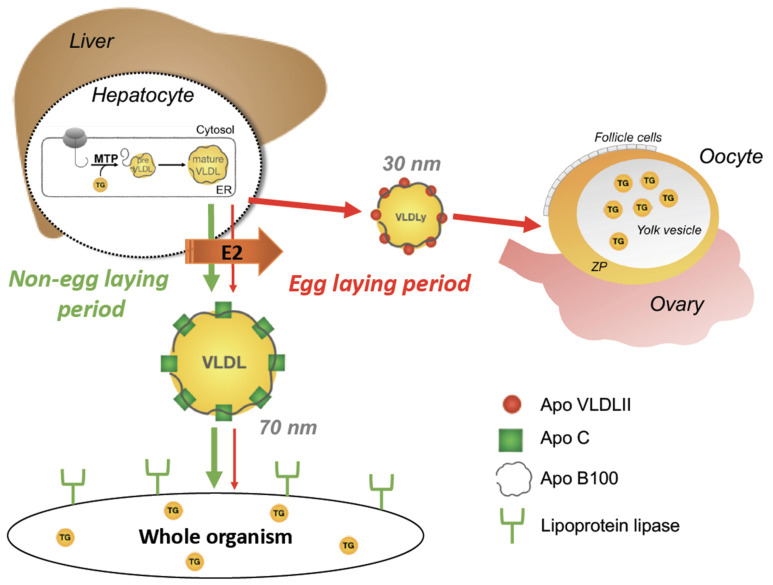
Role of estrogens in lipoprotein metabolism in oviparous vertebrates. VLDL particles (70 nm) produced in the liver provide TGs to the whole body (green arrows). During the egg-laying period, there is a shift induced by E2 to produce lipoprotein lipase-resistant yolk targeted VLDLy particles (30 nm) delivering TGs to the growing oocytes (thick red arrows), which reduces the levels of the 70 nm particles (thin red arrows) during this period. See text for additional information. E2: 17β-estradiol; TG: triglyceride; VLDL: very low density lipoprotein; VLDLy: yolk targeted VLDL; MTP: microsomal triglyceride transfer protein.

**Figure 5 metabolites-11-00502-f005:**
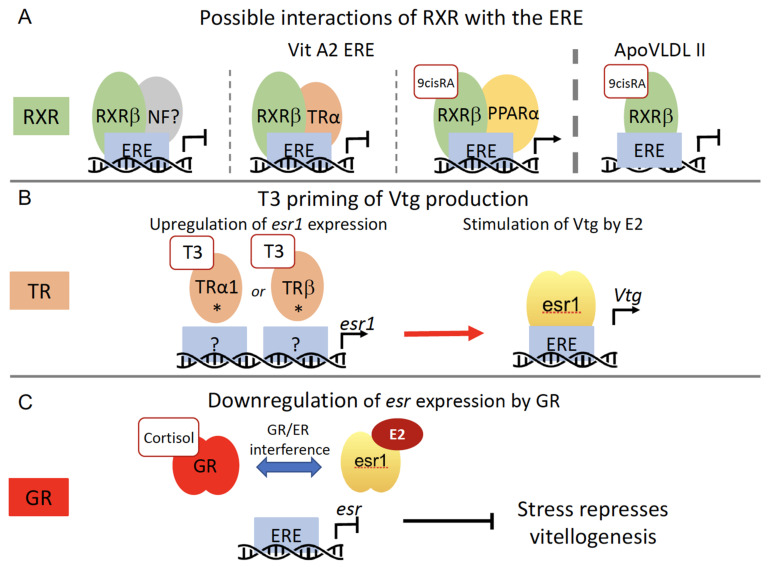
Examples of hepatic crosstalk between esr and other members of the nuclear receptors family in oviparous vertebrates. (**A**) Possible interactions of RXR with the ERE. (**B**) T3 priming of Vtg production. * Indicates that the molecular mechanism by which TR stimulates *esr1* remains to be elucidated [126]. (**C**) Down-regulation of esr expression by GR leading to a repression of vitellogenin synthesis in stress situations [126]. E2: 17β-estradiol; ERE: estrogen responsive element; Vtg: vitellogenin; RXR: retinoid X receptor; NF: nuclear factor; TR: thyroid receptor; PPARα: peroxisome proliferator-activated receptor α; 9cisRA: 9-cis-retinoic acid; T3: triiodothyronine; esr: estrogen receptor; GR: glucocorticoid receptor. 
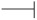
 inhibition; 
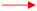
 stimulation.

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
