# Peer review of "Roles of Estrogens in the Healthy and Diseased Oviparous Vertebrate Liver"

_metabolites, 2021, doi:10.3390/metabo11080502_

Round 1

Reviewer 1 Report

The manuscript by Tramunt and co-authors is an interesting and well-organized review on the involvement of liver and estrogen in oviparous vertebrates reproduction. Nevertheless, some points should be deepened:

  • although the authors declare that they do not discuss only the oviparous reproduction of monotremes, in the review the authors focus a lot on fish and amphibians, citing only partially the birds and neglecting the reptiles;
  • in paragraph 7, they describe how the liver of fish (called “lower vertebrates”, thus using an evolutionarily incorrect expression) acts as a sensor towards estrogen disrupting compounds, neglecting the fact that the estrogen-induced synthesis of VTG in the liver of males of terrestrial oviparous vertebrates represent an excellent biomarker of soil contamination by xenoestrogens, as demonstrated by the robust literature on the subject.

in line 52, Kuppfer should be Kupffer

Author Response

Reviewer:

The manuscript by Tramunt and co-authors is an interesting and well-organized review on the involvement of liver and estrogen in oviparous vertebrates reproduction.

Reply:

We thank the reviewer for his/her positive and constructive evaluation of our manuscript, which is much appreciated.

Reviewer:

Nevertheless, some points should be deepened:  

although the authors declare that they do not discuss only the oviparous reproduction of monotremes, in the review the authors focus a lot on fish and amphibians, citing only partially the birds and neglecting the reptiles;

Reply:

We choose to review non-mammalian oviparous vertebrates and, therefore, did not discuss the monotremes that are a group of highly specialised egg-laying mammals. We hope that the reviewer can accept this choice.

The information relevant to the different Chapters and the matter we wanted to address in the review is more or less important for each group of animals, which may give the impression of a biased focus. Concerning birds, the information is the most abundant for these animals, compared to the other groups of animals, on liver lipid metabolism and liver diseases. Thus bird work has a good place in these chapters, and indeed less in other chapters. We agree that reptiles were not much discussed and this has been corrected in the revised version. We thank the reviewer for this constructive criticism. For instance, there is now a whole paragraph on reptiles in Chapter 7, and they are mentioned in Chapter 2 (lines 223-225), and Chapter 3 (lines 279-281).

Reviewer:

In paragraph 7, they describe how the liver of fish (called “lower vertebrates”, thus using an evolutionarily incorrect expression) acts as a sensor towards estrogen disrupting compounds, neglecting the fact that the estrogen-induced synthesis of VTG in the liver of males of terrestrial oviparous vertebrates represent an excellent biomarker of soil contamination by xenoestrogens, as demonstrated by the robust literature on the subject.

Reply:

Our definition of “lower vertebrates” was somewhat broader than that usually used by zoologists. We used it to cover birds, reptiles, amphibians and fish -in other words, the non- mammalian vertebrates- whereas the conventional approach is to include birds with the mammals as 'higher' vertebrates. To avoid confusion, we have now removed the wording “lower vertebrates” and replaced it by “non-mammalian oviparous vertebrates”.

We agree with the reviewer we neglected to mention work done with male terrestrial oviparous vertebrates, especially reptiles, to assess soil contamination by xenoestrogens. This gap has been filled in the revised version (see Chapter 7, lines 616-626). 

Reviewer:

in line 52, Kuppfer should be Kupffer

Reply: Thank you, we corrected.

Reviewer 2 Report

This is an interesting article, detailed description of liver function, vitellogenesis and estrogen regulation in oviparous vertebrates. Onli minor modifications are needed before publishing.

1.In Abstract use ESR instead “esr” and GPER1 instead “gper1” as it is general information not related to nonmammalian vertebrates I guess

  1. Lines 27 and 44 say different information on NAFLD/NASH
  2. Information on estrogen related receptors (ERRs) especially in the context of high energy needed processes should be included e. g. maybe in context of Danio rerio?

3.“Furthermore,we use roman letters to designate proteins and italics for genes.” - it is well known standard

  1. Figure 2 Capture “Distribution” should be changed to “Presence” as the Authors do not inform where exactly about subcellular distribution

Author Response

Reviewer:

This is an interesting article, detailed description of liver function, vitellogenesis and estrogen regulation in oviparous vertebrates. Only minor modifications are needed before publishing.

Reply:

We thank the reviewer for her/his careful evaluation of our manuscript and his/her constructive comments.

Reviewer

  1. In Abstract use ESR instead “esr” and GPER1 instead “gper1” as it is general information not related to nonmammalian vertebrates I guess

Reply:

We have followed the reviewer’s recommendation and are now using ESR and GPER1 in the Abstract.

Reviewer:

  1. Lines 27 and 44 say different information on NAFLD/NASH

Reply:

We have modified the text of Line 27 to solve the question raised by the reviewer.

Reviewer:

  1. Information on estrogen related receptors (ERRs) especially in the context of high energy needed processes should be included e. g. maybe in context of Danio rerio?

Reply:

We thank the reviewer for this suggestion and have added a paragraph at the beginning of Chapter 5 to provide useful information on the role of ERRs in energy homeostasis.

3.“Furthermore,we use roman letters to designate proteins and italics for genes.” - it is well known standard

Reply:

This sentence was removed.

  1. Figure 2 Capture “Distribution” should be changed to “Presence” as the Authors do not inform where exactly about subcellular distribution

Reply:

In Figure 2 Capture and text we have replaced “distribution” by “occurrence”.
